# Factors Determining the Level of Acceptance of Illness and Satisfaction with Life in Patients with Cancer

**DOI:** 10.3390/healthcare11081168

**Published:** 2023-04-19

**Authors:** Renata Piotrkowska, Agnieszka Kruk, Aneta Krzemińska, Wioletta Mędrzycka-Dąbrowska, Katarzyna Kwiecień-Jaguś

**Affiliations:** 1Department of Surgical Nursing, Medical University of Gdansk, Dębinki 7, 80-211 Gdańsk, Poland; agnieszka.kruk@gumed.edu.pl (A.K.);; 2Department of Anaesthesiology Nursing and Intensive Care, Medical University of Gdansk, Dębinki 7, 80-211 Gdańsk, Poland; wioletta.medrzycka-dabrowska@gumed.edu.pl (W.M.-D.); katarzyna.kwiecien-jagus@gumed.edu.pl (K.K.-J.)

**Keywords:** satisfaction with life, quality of life, acceptance of illness, cancer

## Abstract

Introduction: Cancer threatens life and brings about many negative emotions in patients, which influence their satisfaction with life and contribute to a low level of their acceptance of illness. This is why the acceptance of illness is a serious problem among patients with cancer; contributes to the intensification of symptoms; and influences the patient’s physical, mental, emotional, social, and spiritual condition. Aim: The purpose of this work is to assess the acceptance of illness and satisfaction with life in patients with cancer, as well as to identify social, demographical, and clinical factors that significantly differentiate their acceptance of illness and satisfaction with life. Materials and Methods: The study involved 120 patients with cancer aged 18 to 88. The study was conducted in the form of a questionnaire based on standard research tools: Acceptance of Illness (AIS), Satisfaction with Life Scale (SWLS), and Numerical Rating Scale (NRS). Social, demographical, and clinical data were collected in the original questionnaire. Results: A group of 120 patients was studied, including 55.83% (*n* = 67) women and 44.16% (*n* = 53) men. The average age was 56. A general acceptance-of-illness index obtained by the patients was 21.6 ± 7.32 and a general satisfaction-with-life index was 19.14 ± 5.78. The statistical analysis indicated a significant correlation between the acceptance of illness and the intensity of pain (rHO = −0.19; *p* < 0.05), fatigue ((*Z =* 1.92; *p* > 0.05), and diarrhoea (*t_(118)_* = 2.54; *p* < 0.05). The correlation between the intensity of pain and satisfaction with life was negative (rHO = −0.20; *p* < 0.05). Conclusion: The greater acceptance of illness, the greater satisfaction with life in patients with cancer. Pain, fatigue, and diarrhoea decrease the acceptance of illness. In addition, pain decreases the level of satisfaction with life. Social and demographical factors do not determine the level of acceptance of illness and satisfaction with life.

## 1. Introduction

### 1.1. Cancer: Theoretical Aspect

Malignant tumours are the second leading cause for death in Poland. In 2019, they contributed to 25.7% of deaths among men and 23.3% of deaths among women. They constitute a serious health problem, in particular in young and middle-aged people (25–64) [1]. Morbidity and mortality rates in Poland increase mainly as a result of the structure of the aging Polish population, as well as changes connected with exposure to carcinogenic factors, in particular those related to smoking [2]. The report “Health at a Glance” is a comprehensive look at health care systems in different countries around the world. The latest edition of the study, dedicated to countries belonging to the Organization for Economic Co-operation and Development (OECD) and OECD partner countries, was published on 9 November 2021 and includes, among others, updated data on cancer morbidity and mortality. The “Health at a Glance 2021” report indicates that the cancer morbidity rate in Poland is relatively low. In Poland, the average rate is 267 per 100,000 inhabitants, while in other OECD countries, the average rate is 294. However, the cancer mortality rate of 228 deaths per 100,000 inhabitants in Poland is one of greatest among OECD countries in comparison with the average rate in OECD countries of 191 deaths per 100,000. Early diagnosis allows for implementing treatment faster, which increases the survival rate. This is why the mortality rate in Australia and New Zealand is below the average, despite having the greatest morbidity rates. The data indicate that morbidity and mortality rates are greater in men than women in all OECD member countries and partner states [3].

### 1.2. Satisfaction with Life and Acceptance of Illness: Their Meaning in Medicine

The issue of satisfaction with life is strictly connected with quality of life and first appeared in medicine in the 1970s. For several decades, we have been observing changes in the meaning and assessment of quality of life. The World Health Organisation defines health not only as the total lack of illness or disability, but as full physical, mental, and social welfare as well [4]. In medicine, many attempts have been made to define quality of life and no clear definition has been agreed on to date. The most accurate definition of quality of life stipulates that this is an image of a patient’s mental, physical, social, and economic condition as perceived by that patient at the defined time [5]. Quality of life in patients with cancer is assessed using several levels considering their physical, mental, social, and somatic zone [6]. Another indicator that a patient has adapted to illness is acceptance, which means “accepting a judgement, opinion, point of view or way of conduct, having favourable attitudes, or giving consent to something”. The ability to accept can involve many life factors, such as external appearance, life aspirations, intellect, character and self-evaluation [7]. The acceptance of illness, as well as pain and discomfort that are strictly connected with that illness is a variable parameter, which is often applied in the assessment of the patient’s adjustment to illness. Patients with cancer learn to deal with their illness, recognise its symptoms, and feel the impact of that illness on their quality of their life day by day. Every day, they experience discomfort resulting from the lack of independence and a partial or complete change from their previous role in the social life [8]. The acceptance of illness is a variable that reliably reflects quality of life when the patients deals with the illness, which allows for the assessment of the level of satisfaction with life and the present health condition in the holistic context [9].

In accordance with the above assumptions, we propose that when dealing with a life-threatening event, namely cancer, there are related differences in the processes of adaptation and cognition of the disease. Such a discovery may have practical significance. Therapeutic process in such a population should include an evaluation of disease acceptance, as it may allow for the identification of patients with poor acceptance of disease, and assist with planning therapeutic, prophylactic, and educational actions for them.

### 1.3. Aim

The purpose of this work is to assess the acceptance of illness and satisfaction with life in patients with cancer, as well as to identify social, demographical, and clinical factors that significantly differentiate their acceptance of illness and satisfaction with life.

## 2. Materials and Methods

### 2.1. Design

A cross-sectional survey study design was used.

### 2.2. Study Procedures

The study was conducted in the Clinic of Oncology and Radiotherapy of the University Clinical Centre in Gdańsk in 2022. The study involved 120 cancer patients of 18 to 88 years old.

Patient eligibility criteria:

Patient’s voluntary and informed consent;

A diagnosed cancer; 

Age ≥ 18;

Ability to establish a verbal logical contact;

Hospitalisation in the oncology ward as at the recruitment date.

Patient exclusion criteria:

Lack of patient’s consent; 

Age ≤ 18;

An inability to establish a verbal logical contact.

The respondents filled in their questionnaires in a private place during their hospitalisation.

### 2.3. Questionnaire Development

#### 2.3.1. Acceptance of Illness 

The AIS by Felton, Revensson, and Hinrichsen of the Center for Community Research and Acton, Department of Psychology, New York University (adapted to Polish conditions by Juczyński), contains eight statements describing the negative consequences of bad health, including the assessment of limitations imposed by the illness, the lack of self-sufficiency, the sense of dependence on others, and reduced self-esteem. In each statement, the patient defines their present condition on a five-grade scale: from 1 “strongly agree” to 5 “strongly disagree”. The strong agreement (grade 1) reflects a bad adjustment to illness and the strong disagreement (grade 5) reflects the acceptance of illness. A general measure of the acceptance of illness is the total score, which may be from 8 to 40. Results below 20 are considered low and reflect the lack or poor acceptance of and adjustment to illness, as well as significant emotional problems related to illness. Results from 20 to 30 reflect moderate acceptance. Results above 30 reflect high or full acceptance [10]. The reliability of the Polish version is satisfactory and Cronbach’s alpha is 0.85 [11].

#### 2.3.2. The Satisfaction with Life Scale 

The SWLS by Diener, Emmons, Larsen, and Griffin of the Department of Psychology, University of Illinois (adapted to Polish conditions by Juczyński), assesses satisfaction with life. It is a short method made of five items assessed through the use of a seven-point scale. The respondent is requested to take a position on each statement by specifying to which extent each of those statements reflects their life, from strongly agree (7 points) to strongly disagree (1 point). The scores are summed up and the total result reflects the patient’s satisfaction with life. The results range between 5 and 35. The greater the result, the greater the satisfaction with life. Psychometric properties of the SWLS are satisfactory. In the original version, the reliability index α was 0.87 [10,12]. The reliability of the Polish version SWLS is satisfactory, and Cronbach’s alpha is 0.82 [10].

#### 2.3.3. Numerical Rating Scale 

The NRS is a numerical scale of 11 pain intensity degrees: from 0 to 10, where 0 means a complete lack of pain and 10 is the worst pain felt by the patient. An NRS result of up to 3 means a mild pain, 4–6 moderate pain, and 7–10 the strongest imaginable pain [13,14].

#### 2.3.4. Original Questionnaire

The questionnaire allowed for the collection of social and demographic data, such as sex, age, education level, place of residence, marital status, and professional activeness. Other questions referred to the history of illness include clinical diagnosis, duration, type of treatment, and illness-related symptoms.

### 2.4. Ethical Considerations

The data were collected in accordance with ethical principles set out in the Declaration of Helsinki. The survey was conducted based on consent given by the Independent Bioethical Commission of the Medical University in Gdańsk, No. NKBBN/163/2022.

### 2.5. Statistical Analysis

All statistical calculations were made using the IBM SPSS 23 statistical package and Excel 2016 spreadsheets. Qualitative variables were presented in the form of sizes and percentage values, and quantitative variables were presented in the form of an arithmetic mean and standard deviation. The significance of differences between more than two groups was subject to a Kruskal–Wallis test (if significant differences were obtained, post-hoc Bonferroni tests were made) and between two groups to a Mann–Whitney U test. To confirm a relationship between a force and a direction of the variables, a correlation analysis was made and Spearman’s correlation coefficients were calculated. In all calculations, a significance level of *p* ≤ 0.05 was assumed.

## 3. Results

### 3.1. Social and Demographic Characteristics of the Respondents

A group of 120 patients was studied, including 55.83% (*n* = 67) women and 44.16% (*n* = 53) men. Patients were 18 to 88 years old. The average age was 56. Moreover, 64.16% (*n* = 77) of the respondents stated that they came from a city and 35.83% (*n* = 43) from a village. Patients with a secondary education constituted the largest group at 35% (*n* = 42), followed by patients with vocational education at 30.83% (*n =* 37), higher education at 25% (*n* = 30), and primary education at 9.16% (*n* = 11). Half of patients were married (50.0%). Moreover, 39.16% (*n* = 47) were professionally active and 34.16% (*n* = 41) were retired. Most patients assessed their financial status as being good (61.7%). 

### 3.2. Clinical Characteristics of the Group

The most frequent cancers in the group of respondents were as follows: lung cancer (24.16%), breast cancer (16.66%), and reproductive organ cancer (15%). Over a half of the respondents (62.50%, *n* = 75) confirmed that there was cancer in their family. Almost a half of the patients stated that they had had cancer for up to 6 months (45%) and 39.16% ha for more 6 months to 1 year. Furthermore, 49.16% of patients responded that they had been diagnosed with cancer from 1 to 3 months ago. Moreover, 80.83% of patients were undergoing chemotherapy and 65% radiotherapy. During their oncological treatment, the patients most frequently suffered from diarrhoea (66.66%, *n* = 80), pain (53.33%, *n* = 64), and fatigue (29.16%, *n* = 35), as shown in Table 1. An average pain intensity based on the NRS was 5.24 ± 2.37, as shown in Table 2.

### 3.3. Assessment of Satisfaction with Life

The statements “in most ways my life is close to my ideal” and “the conditions of my life are excellent” were scored close to the “strongly disagree” assessment. In the case of statements “I am satisfied with my life” and “so far I have gotten the important things I want in life ”, the average scoring was from 4.06 ± 1.56 to 4.14 ± 1.67, which was the closest to “strongly agree”. The statement “if I could live my life over, I would change almost nothing” was assessed in the most optimistic way: the average of 4.26 ± 1.85, which means that most respondents agreed with that sentence, as shown in Table 3. A general satisfaction-with-life index was 19.14 ± 5.78. The greater the index, the greater satisfaction with life, as shown in Table 4.

### 3.4. Assessment of Acceptance of Illness

The results indicate that cancer patients had a problem adjusting to the limitations imposed by their illness (2.65 ± 1.20), could not do what they enjoyed most (2.32 ± 1.13), and were more dependent than they would like to have been because of their health condition (2.63 ± 1.25). This means that the acceptance of cancer in these areas was low. The statement “illness makes me feel unnecessary sometimes” (2.8 ± 1.22) obtained the highest result. The statements “illness makes me a burden to my family and friends” (3.01 ± 1.33) and “my state of health makes me not feel like a full-fledged human” (3.02 ± 1.38) obtained the greatest average values, which means that the patients and their relatives better accepted the consequences of cancer, as shown in Table 5. The general acceptance-of-illness index obtained by the patients was 21.6 ± 7.32, which reflected the average level of acceptance of cancer, as shown in Table 6. 

### 3.5. Social and Demographic Factors and the Level of Acceptance of Illness and Satisfaction with Life

The age, sex, and place of residence of the respondents did not have a great impact on the level of acceptance of illness and satisfaction with life, as shown in Table 7 and Table 8.

### 3.6. Satisfaction with Life Versus Acceptance of Illness

To verify the hypothesis, the Spearman correlation test was conducted. Close to the limit of a statistical tendency, a positive correlation between the variables was obtained. This means that the acceptance of illness increased along with the growth in satisfaction with life (rHO = 0.17; *p* = 0.056), as shown in Table 9, Figure 1.

### 3.7. Acceptance of Illness versus Duration of Illness, Pain Intensity, and Other Symptoms of Cancer

AIS

Duration of illness

To verify the hypothesis, the Spearman correlation test was conducted. No statistically significant correlation between variables was obtained (rHO = −0.14; *p >* 0.05). The duration of illness did not affect the acceptance of illness. 

Pain intensity 

To verify the hypothesis, the Spearman correlation test was conducted. There was a negative correlation between the variables. This means that the acceptance of illness decreased along with the increase in pain (rHO = −0.19; *p* < 0.05), as shown in Table 10, Figure 2.

Disease-related symptoms

Student’s T tests were conducted for independent samples and Mann–Whitney samples. The analysis indicated a relationship between the variables. A statistically significantly smaller acceptance of illness was recorded in the case of the respondents suffering from diarrhoea (t_(118)_ = 2.54; *p* < 0.05) and fatigue (Z = 1.92; *p* > 0.05). No statistically significant differences were recorded between the acceptance of illness and the existence of symptoms: nausea (Z = 0.23; *p* > 0.05), constipation (Z = 0.96; *p* > 0.05), pain (t_(118)_ = 0.14; *p >* 0.05), and dyspnoea (Z = 0.48; *p >* 0.05), as shown in Table 11. 

### 3.8. Satisfaction with Life versus Duration of Illness, Pain Intensity, and Disease-Related Symptoms

SWLS

Duration of illness and disease-related symptoms

The duration of illness (*p* = 0.437) and disease-related symptoms (nausea (Z = 0.21; *p* > 0.05), constipation (Z = 0.38; *p* > 0.05), diarrhoea (*t_(118)_* = 1.25; *p* > 0,05), fatigue (Z = 0.32; *p* > 0.05), pain (*t_(118)_* = 0.85; *p* > 0.05), and dyspnoea (Z = 1.33; *p* > 0.05)) do not have a material impact on satisfaction with life.

Pain intensity 

To verify the hypothesis, the Spearman correlation test was made. There is a negative correlation between the variables. This means that the satisfaction with life decreases along with the growth of pain (rHO = −0.20; *p <* 0.05), as shown in Table 12, Figure 3.

## 4. Discussion

In accordance with epidemiological forecasts from the World Health Organisation, the global cancer burden will increase to 30 million by 2040 [15]. Therefore, the demand for related medical care is increasing and scientists need to conduct studies on oncology [15]. 

The results of the existing studies confirm that a greater acceptance of cancer leads to less suffering, as well as fewer depression and anxiety symptoms [16]. In addition, there is a positive relationship between the acceptance of cancer and the high mental and physical quality of life and functioning of cancer patients [17,18,19]. The acceptance of illness can reflect adaptation to the illness [20,21,22]. Researchers have proven that patients with a low acceptance of illness are more prone to negative emotions and poorer adjustment, and tend to withdraw from recommended cancer therapies more frequently [20]. As a result of mistaken beliefs, the lack of acceptance of illness is frequently a greater problem than the illness [23].

Analysing the level of acceptance of illness in our own studies, we found that the level of acceptance of illness in patients was average (21.6). The patients obtained the greatest result with regard to the statement “my state of health makes me not feel like a full-fledged human” (3.02). The criterion “illness makes me a burden to my family and friends” obtained a slightly smaller acceptance (3.01). The response “due to my state of health, I am not able to do what I like the most” (2.32) obtained the smallest score in the AIS. In the study conducted by Branecka-Woźniak with partners among 145 cancer patients, the acceptance of illness was at an average level [24]. The results of studies conducted by Smoleń and partners in the group of 229 patients with cancer confirmed that the greatest number of patients had a medium and high level of acceptance of cancer, similarly to the research of Czerw and partners, where patients with lung cancer had an average level of acceptance [25]. Such an approach to illness may have a positive impact on treatment and improve the quality of life [7]. The acceptance of illness was slightly above the average in the research of Krajewski and partners (28.8) and Lewandowska-Abucewicz and partners (26.8), which indicates that the acceptance of cancer was good [26,27]. In turn, in the study conducted by Kozera, the acceptance of illness in the target group was low, which had an impact on the sense of security [28].

Analysing the level of satisfaction with life in our own study, we can state that the patients achieved an average result in the SWLS (19.14). In our study, the statement “if I could live my life over, I would change almost nothing” obtained the highest score in SWLS (4.26). A similar score was obtained by the statements: “So far I have gotten the important things I want in life” (4.14) and “I am satisfied with my life” (4.06). The results obtained by Spanish researchers in the group of 713 patients confirmed a greater satisfaction with life with an average level of 27.1 in the SWL scale [29].

Our own study indicated a relationship between the acceptance of illness and satisfaction with life in cancer patients. Many studies on various types of cancer indicate that the acceptance of illness improves the quality of all areas of life and the satisfaction with life [30,31].

At present, the acceptance of illness is considered as an independent predictor of a decreased quality of life in many chronic diseases [28]. In turn, the acceptance of illness and satisfaction with life can be determined by a number of social, demographic, and clinic factors. In our own study, the acceptance of illness and satisfaction with life were not dependent on such factors, for example, age, sex, and place of residence. In other studies [7,32,33], no impact of sex and place of residence on the acceptance of illness was recorded, and the acceptance was not found to be dependent on age and place of residence either [25]. In turn, in the study conducted by Czerw and partners [34], social and economic factors, such as education, place of residence, income, and professional status, have an impact on the acceptance of illness by patients. In the studies of Kołpa and partners [35], the acceptance of illness was only dependent on age. Summing up, it can be stated that there is no direct relation between the acceptance of illness and satisfaction with life and the patient’s age, sex, education, and place of residence. The age, sex, education, and place of residence can be assumed to modify those variables indirectly.

In this study, patients suffering pain, tiredness, and diarrhoea had a worse acceptance of illness. In addition, pain intensity also reduced the patients’ satisfaction with life. Patients with cancer with a greater number of complaints related to their illness reported greater mental exhaustion and a worse acceptance of illness [36]. The researchers indicated that the following symptoms were most irritating to the patients, pain, diarrhoea, and constipation, which was also reflected in our own studies [37].

Treating pain complaints is a very important aspect, because studies indicate that this symptom significantly decreases their acceptance of illness and satisfaction with life. Cancer pain is usually long-lasting and changes into chronic pain. The patients feel depressed, are frequently physically exhausted, and these symptoms take control over their life. Family and friends also play an important role as they take on new functions and duties, but they should also be involved in cancer pain therapy. The patients should be open to support from other people and foundations, which can offer a wide range of aid aimed at improving the quality of life for cancer patients [38].

Fatigue is a frequent problem for patients with cancer. Cancer fatigue symptom is connected both with the illness itself, as well as treatment methods, and can affect physical, mental, and social functioning. This problem is often underestimated both by patients and medical professionals. It is important to counteract fatigue, because it has a significant impact on the acceptance of illness by patients and on their quality of life [39,40]. Diarrhoea as a consequence of cancer and related treatment methods may affect the success of cancer therapy and contribute to the extension or suspension of treatment. It also decreases satisfaction with life [41]. In our study, the patients suffering from diarrhoea had a worse acceptance of disease.

## 5. Limitations

The limitation of this study is the low representativeness of the results, which makes it difficult to generalize the obtained results and formulate broader conclusions. The subject of this study was limited to patients hospitalized in one medical center in Gdańsk. In subsequent studies, the number of patients tested should be increased. We plan to conduct research using this methodology in other regions of Poland in order to make it a multicenter study and acceptable as a national crossectional study.

## 6. Conclusions

The greater the acceptance of illness, the greater the satisfaction with life in patients with cancer. Pain, fatigue, and diarrhoea decrease the acceptance of illness. In addition, pain decreases the level of satisfaction with life. Social and demographical factors do not determine the level of acceptance of illness and satisfaction with life.

This study may help staff recognize the needs and determine the extent of care and education needed for patients with cancer so they feel safe during the treatment of cancer.

### Implications for Use in Practice

The results can be useful to understand the impact of certain social, demographic and clinic factors on the assessment of the acceptance of illness and satisfaction with life. The monitoring and effective handling of pain, fatigue, and diarrhoea can improve the acceptance of illness substantially. Taking advantage of other specialised support, e.g., psychological, rehabilitation, and care services, also contributes to an improvement in the acceptance of illness and in the adjustment to the new circumstances of life. In clinical practice, actions must be taken to assess the acceptance of illness in order to identify patients with a low acceptance and plan relevant treatment, preventive, and educational actions.

## Figures and Tables

**Figure 1 healthcare-11-01168-f001:**
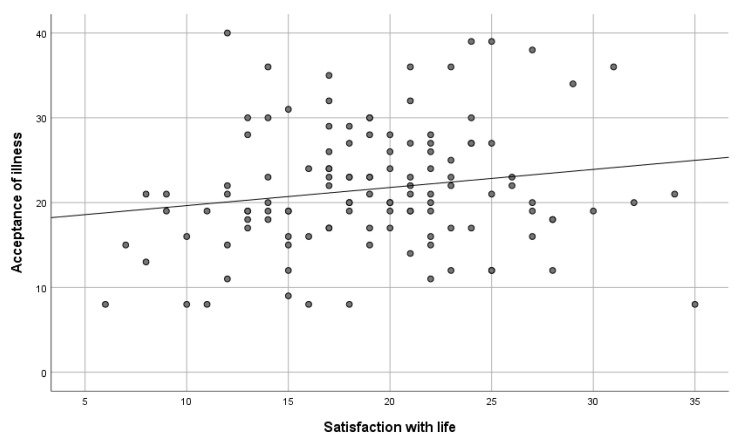
Satisfaction with life versus acceptance of illness.

**Figure 2 healthcare-11-01168-f002:**
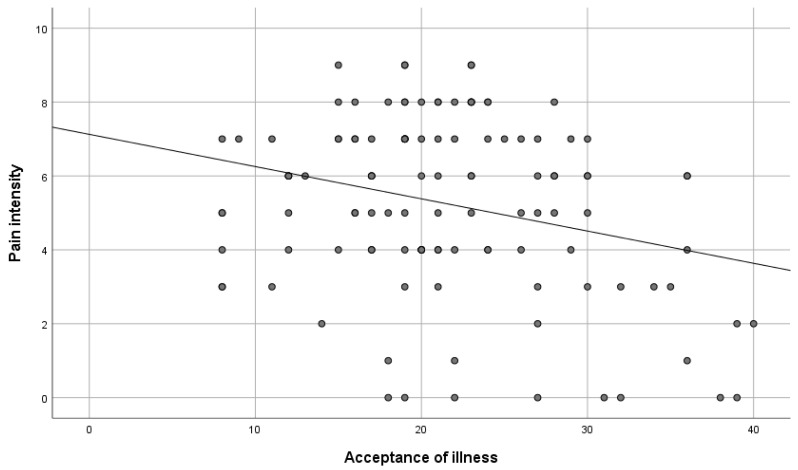
Pain intensity versus acceptance of illness.

**Figure 3 healthcare-11-01168-f003:**
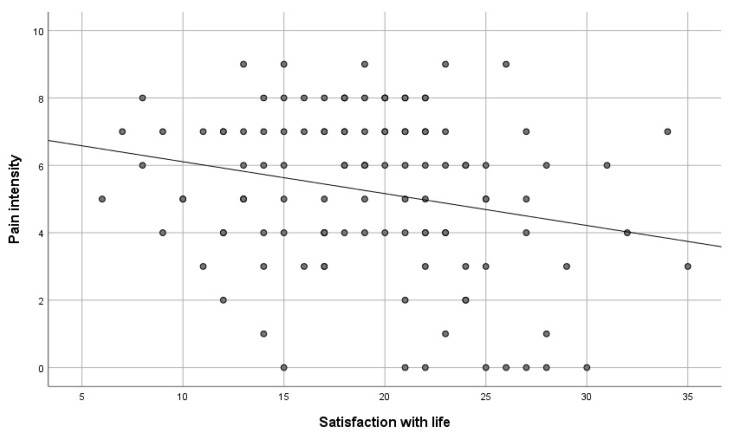
Pain intensity versus satisfaction with life.

**Table 1 healthcare-11-01168-t001:** Clinical characteristics of the studied group.

	N = 120	%
Have been cancers in your family?		
Yes	75	62.50
No	45	37.50
Duration of cancer		
Up to 6 months	54	45.00
Over 6 months to 1 year	47	39.16
Over 3 years	19	15.83
Type of cancer		
Breast cancer	20	16.66
Colorectal cancer	13	10.83
Bone cancer and multiple myeloma	3	2.50
Gastrointestinal system cancer	8	6.66
Skin cancer	10	8.33
Lung cancer	29	24.16
Vascular system cancer	2	1.66
Urinary tract cancer	8	6.66
Reproductive organs cancer	18	15.00
Other	9	7.50
Applied type of treatment		
Chemotherapy	97	80.83
Radiotherapy	78	65.00
Surgical treatment	57	47.50
Immunotherapy	7	5.83
Hormonotherapy	14	11.66
Symptoms during treatment of cancer		
Nausea	31	25.83
Constipation	28	23.33
Diarrhoea	80	66.66
Fatigue	35	29.16
Pain	64	53.33
Dyspnoea	29	24.16
Other	63	52.50

N—number of respondents.

**Table 2 healthcare-11-01168-t002:** Pain intensity at NRS.

	N	Min	Max	M	SD
Pain intensity at NRS	120	0	9	5.24	2.37

N—number of respondents; M—mean value; SD—standard deviation.

**Table 3 healthcare-11-01168-t003:** Satisfaction with life assessed using the SWLS questionnaire.

	N	Min	Max	M	SD
In most ways my life is close to my ideal	120	1	7	3.32	1.34
The conditions of my life are excellent	120	1	7	3.37	1.54
I am satisfied with my life	120	1	7	4.06	1.56
So far I have gotten the important things I want in life	120	1	7	4.14	1.67
If I could live my life over, I would change almost nothing	120	1	7	4.26	1.85

N—number of respondents; M—mean value; SD—standard deviation.

**Table 4 healthcare-11-01168-t004:** General SWLS index.

	N	Min	Max	M	SD
SWLS	120	6	35	19.14	5.8

N—number of respondents; M—mean value; SD—standard deviation.

**Table 5 healthcare-11-01168-t005:** Acceptance of the illness according to the AIS questionnaire.

Descriptive Statistics (Alfa Cronbacha = 0.87)	N	Min	Max	M	SD
I have trouble adjusting to the limits imposed by the disease	120	1	5	2.65	1.20
Due to my state of health, I am unable to do what I like the most	120	1	5	2.32	1.13
Illness makes me feel unnecessary sometimes	120	1	5	2.80	1.22
Health problems make me more dependent on others than I want to be	120	1	5	2.63	1.25
Illness makes me a burden to my family and friends	120	1	5	3.01	1.33
My state of health makes me not feel like a full-fledged human	120	1	5	3.02	1.38
I will never be self-sufficient to the extent that I would like to be	120	1	5	2.68	1.28
I think people who are with me are often embarrassed by my illness	120	1	5	2.50	1.22

N—number of respondents; M—mean value; SD—standard deviation.

**Table 6 healthcare-11-01168-t006:** General acceptance-of-illness index.

	N	Min	Max	M	SD
AIS	120	8	40	21.60	7.32

N—number of respondents; M—mean value; SD—standard deviation.

**Table 7 healthcare-11-01168-t007:** Acceptance of illness and satisfaction with life versus age.

AIS	N	rHO	*p*
Age	120	−0.04	0.659
SWLS	N	rHO	*p*
Age	120	−0.01	0.882

N—number of respondents; rHO—Spearman’s rank correlation coefficient; *p*—level of significance.

**Table 8 healthcare-11-01168-t008:** Acceptance of illness and satisfaction with life versus sex and place of residence.

**AIS vs. Sex**	**N**	**M**	**SD**	**t**	**df**	** *p* **
Woman Men	67 53	21.71 21.45	7.92 6.54	0.19	118	0.846
**AIS vs. place of residence**	**N**	**M**	**SD**	**t**	**df**	** *p* **
Town Village	77 43	22.11 20.67	7.24 7.45	1.03	118	0.303
**SWLS vs. sex**	**N**	**M**	**SD**	**t**	**df**	** *p* **
Woman Men	67 53	19.40 18.81	6.62 4.54	0.55	118	0.580
**SWLS vs. place of residence**	**N**	**M**	**SD**	**t**	**df**	** *p* **
Town Village	77 43	19.68 18.16	5.70 5.87	1.39	118	0.167

N—number of respondents; M—mean value; SD—standard deviation; df—degrees of freedom; t—the result of Student’s *t*-test; *p*—level of significance.

**Table 9 healthcare-11-01168-t009:** Satisfaction with life versus acceptance of illness.

AIS	N	rHO	*p*
SWLS	120	0.17	0.056

N—number of respondents; rHO—Spearman’s rank correlation coefficient; *p*—level of significance.

**Table 10 healthcare-11-01168-t010:** Acceptance of illness versus pain intensity.

AIS	N	rHO	*p*
Pain (NRS)	120	−0.19	0.049

N—number of respondents; rHO—Spearman’s rank correlation coefficient; *p*—level of significance.

**Table 11 healthcare-11-01168-t011:** Acceptance of illness versus disease-related symptoms.

**AIS vs. Nausea**	**N**	**M**	**SD**	**Z**		** *p* **
No Yes	89 31	21.55 21.74	7.50 6.87	0.23		0.817
**AIS vs. constipation**	**N**	**M**	**SD**	**Z**		** *p* **
No Yes	92 28	21.94 20.46	7.59 6.30	0.96		0.334
**AIS vs. diarrhoea**	**N**	**M**	**SD**	**t**	**df**	** *p* **
No Yes	40 80	23.95 20.42	8.87 6.13	2.54	118	0.012
**AIS vs. fatigue**	**N**	**M**	**SD**	**Z**		** *p* **
No Yes	83 35	22.36 19.74	7.32 7.06	1.92		0.050
**AIS vs. dyspnoea**	**N**	**M**	**SD**	**Z**		** *p* **
No Yes	91 29	21.80 20.96	7.50 6.79	0.48		0.625

N—number of respondents; M—mean value; SD—standard deviation; df—degrees of freedom; t—the result of the student’s *t*-test; Z—normal distribution, the result of the Z test; *p*—level of significance.

**Table 12 healthcare-11-01168-t012:** Satisfaction with life versus pain intensity.

AIS	N	rHO	*p*
Pain (NRS)	120	−0.20	0.032

N—number of respondents; rHO—Spearman’s rank correlation coefficient; *p*—level of significance.

## Data Availability

A dataset will be made available upon request to the corresponding authors one year after the publication of this study. The request must include a statistical analysis plan.

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
