# Peer review of "Factors Determining the Level of Acceptance of Illness and Satisfaction with Life in Patients with Cancer"

_healthcare, 2023, doi:10.3390/healthcare11081168_

Round 1

Reviewer 1 Report

Dear Authors,                                                                                                                             the topic of your article is interesting and timely. The design of the study offers many limits due to the cross-sectional study characteristics. Your statistical analysis is good and the results are well presented but the discussion is not argued and adds no significant value to the result. The bibliography is limited and includes too many authors that are your close collegues. I recommend a minor revision focused on improving the discussion and the bibliography to support the strength of the results. Please, revise English language and style. If revised I think it would be of significant value when published. 

Author Response

Review 1

Thank you very much for your positive and constructive feedback. We have tried to take into account all comments and correct them. The authors re-analyzed the study results and revised the conclusions. We added the limitations of our study. Although this is an exploratory study conducted on a limited sample of patients, the results obtained may contribute to expanding knowledge in this area.

  1. Please, revise English language and style. If revised I think it would be of significant value when published. 

        Answer: The work was translated by a professional translation agency with Native Speaker.

  1. The bibliography is limited.

Answer: The bibliography includes researchers from Poland. The authors of these articles are   not my close associates.

  1. The discussion is limited.

Answer: Improved the discussion

Reviewer 2 Report

Factors determining the level of acceptance of illness and satisfaction with life in cancer patients: a national cross-sectional  study

SUMMARY 

The purpose of this study is to determine the level of acceptance of cancer as a disease entity and also to identify the quality and satisfaction with life in cancer patients. Also, the study aims to identify social, demographic, and clinical factors that help to differentiate between the acceptance of cancer as a terminal disease and the quality of life thereafter.

The study concluded that the greater the acceptance of cancer as a disease, the higher the level of satisfaction with life. Pain fatigue and diarrhea are negative factors that decrease the acceptance of the living reality of cancer. This seems very obvious actually 

STRENGTH 

Detailed and concise statistical analysis. Diagram also appropriate 

WEAKNESS 

Small sample size, not enough statistical power. It is difficult to call this a national cross-sectional study 

The disparity in the patient location is 64.16 % from the city and 35.83 from rural settings. Does the urban and rural dwelling have an impact on this study? This might be an important factor 

Is the  AIS by Felton, Revensson, and Hinrichsen of the Center for Community Research 99 and Acton, Department of Psychology, New York University a validated study for acceptance of illness if so, please state it 

Is SWLS by Diener, Emmons, Larsen, Griffin of the Department of Psychology, University of Illinois a well-validated tool for assessment of the Satisfaction of Life Scale, if so please state it 

Please cite where the NRS was derived.

CONCLUSION 

Good work, very detailed but needs some revision. I will also suggest using this methodology and conducting the study in other regions of Poland to make it a multicenter study and acceptable as a national crossectional study

Reviewer 3 Report

This study has been focused towards exploring the Factors determining the level of acceptance of illness and satisfaction with life in cancer patients: a national cross-sectional study. My first impression is that the manuscript is well written, however contain very limited experimental evidences which is based on questionnaire and very broad explanation of content. Authors have described the concept to a greater extent but the quality of manuscript is very low in terms of experimental study. Some major issues suggested here. The manuscript can be considered for publication after careful revision.

·       Elaborate the section 1.1 and correlate it with the title and significance of this study. Also explain the reason behind the formulation of this study.

·       Properly format the study design section. Text is not aligned properly.

·       Please attach the questionnaire format as supplementary file.

·       Please reformat the graph in better quality.

·       Improve the conclusion part by incorporating the content related to future projections.

·       Please aligned the text by interlinking sections with each other.

·       Typographical and grammatical errors are there.

Author Response

Review 3

Thank you very much for your positive and constructive feedback. We have tried to take into account all comments and correct them. The authors re-analyzed the study results and revised the conclusions. We added the limitations of our study. Although this is an exploratory study conducted on a limited sample of patients, the results obtained may contribute to expanding knowledge in this area.

  1. Elaborate the section 1. and correlate it with the title and significance of this study. Also explain the reason behind the formulation of this study.

Answer: In accordance with the above assumptions, we propose that when dealing with a life-threatening event, which is cancer, there are related differences in the processes of adaptation and cognition of the disease. Such a discovery may have practical significance. Therapeutic process in such a population should include an evaluation of disease acceptance, as it may allow the identification of patients with poor acceptance of disease and thus the planning of therapeutic, prophylactic and educational actions for them.

  1. Please attach the questionnaire format as supplementary file.

Answer: corrected

  1. Please reformat the graph in better quality.

Answer: corrected

  1. Please aligned the text by interlinking sections with each other.

Answer: corrected

  1. Typographical and grammatical errors are there.

Answer: corrected

  1. Improve the conclusion part by incorporating the content related to future projections.

Answer: This study may help staff recognize the needs and determine the extent of care and education needed for patients with in cancer patients to feel safe during treatment of cancer.

Reviewer 4 Report

Comments to the author

The authors (Renata et al) have reported developed Atorvastatin-loaded PLGA microparticles using spray drier method for the prevention of intimal hyperplasia Factors determining the level of acceptance of illness and satisfaction with life in cancer patients: a national cross-sectional study. The study is supported with good presentation and data design. However, there are some points which need to be taken care of. Following are some of the comments that the authors might find useful for revised submission. The manuscript should be revised before publication.

1.      Author should mention the fill full of OECD.

2.      What is the meaning of “Descriptive statistics (Alfa Cronbacha = 0,87)”

3.      Author should correct the and similar pattern of writing the values like 2.8±1.22 should 2.8 ± 1.22. it is written as different in different place. Please check in whole manuscript and correct it.

4.      The whole manuscript should be carefully analyzed to get rid of typo errors.

Round 2

Reviewer 2 Report

Thank you for taking some of the suggestions into consideration. Small study but has some scientific merits.

Reviewer 3 Report

Authors have revised the manuscript as per the suggestions thus it can be considered for publication.